# Validation of the MAastricht Instrument of Sustainable Employability (MAISE-NL) Adapted for Employees in Low-Skilled Jobs (MAISE-Easy)

**DOI:** 10.3390/ijerph19137977

**Published:** 2022-06-29

**Authors:** Pauline Mignon, Emmelie Hazelzet, Angelique De Rijk, Hans Bosma, Inge Houkes

**Affiliations:** 1Faculty of Health, Medicine and Life Sciences, Maastricht University, P.O. Box 616, 6200 MD Maastricht, The Netherlands; p.mignon@student.maastrichtuniversity.nl; 2Department of Social Medicine, CAPHRI Care and Public Health Research Institute, Faculty of Health, Medicine and Life Sciences, Maastricht University, P.O. Box 616, 6200 MD Maastricht, The Netherlands; angelique.derijk@maastrichtuniversity.nl (A.D.R.); hans.bosma@maastrichtuniversity.nl (H.B.); inge.houkes@maastrichtuniversity.nl (I.H.)

**Keywords:** sustainable employability, questionnaire, employees in low-skilled jobs, measurement, psychometric properties

## Abstract

*Background*: Sustainable employability (SE) is important for work organizations. Recently, the MAastricht Instrument for Sustainable Employability (MAISE-NL) was developed and validated. This study describes the development and validation of an adapted version of the MAISE-NL, the MAISE-Easy, which can be used for employees in low-skilled jobs. *Methods*: The adaptation of the MAISE-NL was based on six focus groups with employees in low-skilled jobs in various sectors. The MAISE-Easy was distributed among employees in five organizations. The response rate (*n* = 1033) was 53%. Construct validity, reliability and criterion validity were analyzed by means of principal component analysis (PCA), confirmatory factor analysis (CFA), Cronbach’s alpha and correlational analyses. *Results*: The MAISE-Easy included 17 scales divided over four main areas: (1) level of SE; (2) factors affecting SE; (3) overall responsibility for SE; (4) responsibility for factors affecting SE. Construct validity, reliability and criterion validity were adequate to good. *Conclusions*: The MAISE-Easy is a well-validated instrument for measuring SE among employees in low-skilled jobs in terms of the level of SE, factors affecting SE, responsibility for SE and responsibility for factors affecting SE. MAISE-Easy is recommended for both needs assessments and evaluation research in as yet underserved groups of low-skilled workers.

## 1. Introduction

Due to demographic changes and a progressively complex labor market, employers are faced with a graying, overburdened workforce, which increasingly suffers from (chronic) health problems [1]. Therefore, many employers take measures to improve their employees’ sustainable employability (SE). However, there is a lack of scientific consensus on the optimal content of these interventions and a lack of high-quality evaluation studies of SE interventions [2]. Hazelzet and colleagues [2] suggest that effective SE interventions should be better tailored to the needs of both employees and employers and should at least address the four main components of SE that can be deduced from the definition of Van der Klink et al. [3]: health (e.g., physical and mental), productivity (e.g., work ability), valuable work (e.g., meaningful work and positive attitude) and a long-term perspective (e.g., future employability, long-term effects). We consider SE to be a result of an employee–job environment interaction rather than only an individual characteristic. This is also in line with the notion of Van der Klink et al. [3]. Thus, an employee who is healthy, works productively and feels engaged now and in the future, has the positive attitude and competences that fit the job, is sustainably employable.

However, in the development and implementation of SE interventions, the employee perspective is often ignored, even though it is known that employees consider SE to be a shared responsibility between themselves and their employer [4]. The lack of a solid evidence base for SE interventions might relate to a lack of good quality measurement instruments for this concept.

There is an urgent need for a valid SE measurement instrument, which expressly includes the employee’s perspective, is easy to use for researchers, employers and employees, and preferably measures the four core components of SE [5]. SE measurement instruments based on the employees’ perspective currently fall short, particularly for the understudied group of employees in low-skilled jobs. A total of 39% of the Dutch labor population work in this type of jobs [6]. These employees generally have lower levels of education, and their work is often characterized by low levels of job control and high physical demands. Employees in low-skilled jobs have different needs, resilience, skills and knowledge than employees in middle or high-skilled jobs [7,8]. Moreover, they may have a different perspective on SE than higher-skilled employees with higher educational levels [5]. All this may negatively impact the validity of SE measures in this target group and may also explain the low level of response to questionnaires in this group of employees.

In an earlier study, the Maastricht Instrument for Sustainable Employability (MAISE-NL) was developed [5]. The MAISE-NL (which was the basis for the MAISE-Easy) was based on the then available literature on SE and interviews with experts, professionals working in the field and employees. This was a rather explorative process in which we expressly wanted to include the employee perspective. A selection of items was provided to a group of employees, and they were asked about what is important for them with respect to sustainable employability (SE). This process was to a lesser extent guided by theoretical notions on concepts or dimensions related to SE. The measurement instrument resulting from this process (MAISE-NL) was validated first in an explorative factor analysis, and the results were confirmed in the confirmative factor analysis (CFA). These analyses led to two major factors of SE, which were labeled as “productivity” and “health”. The productivity subscale reflects an employee’s ability to be productive, avoid sickness absence, work until retirement and make a decent living. The health subscale of the MAISE-NL reflects an employee’s physical and mental health and the sense of performing meaningful and useful work. The MAISE-NL aimed to address the disadvantages of otherwise valuable existing measurements for SE, such as the capability set for work [9] (highly complex) and the vitality scan [10] (primarily developed from an employer’s and theoretical perspective and validated in an elder and relatively highly educated sample of employees). The MAISE-NL encompasses five main areas: (1) the meaning of SE, (2) the level of SE of the employee, (3) factors affecting SE, (4) the responsibility for SE and (5) the responsibility for factors affecting SE. The MAISE-NL has been tested and validated in samples consisting of middle to highly educated employees and appeared to have good construct validity and reliability [5]. 

### Aim of the Study, Research Questions and Hypotheses

This paper aims to describe the development process of an adapted version of the MAISE-NL for employees in low-skilled jobs, the MAISE-Easy, and to assess the psychometric properties of the MAISE-Easy in terms of construct validity, reliability and criterion validity. 

We hypothesized that the factorial structure of the MAISE-Easy will be confirmed (Hypothesis 1a) and that Cronbach’s alphas of the MAISE-Easy scales will be adequate to good (>0.70) [11] (Hypothesis 1b). 

With regard to the criterion validity, we hypothesized that the level of SE (MAISE-Easy Area 1 (Area 2 in MAISE-NL)) will correlate positively with the criteria vitality and work engagement (Hypothesis 2a). We also hypothesized that the MAISE-Easy Area 1 differentiates between the subgroups regarding gender, age and educational level (all grouped as Hypothesis 2b). We did not formulate specific hypotheses regarding the criterion validity of (responsibility for) factors affecting SE (MAISE-Easy Areas 2, 3 and 4).

## 2. Materials and Methods

### 2.1. Development of the MAISE-Easy for Employees in Low-Skilled Jobs

Using the input of focus groups, the MAISE-Easy consists of the adapted scales from the MAISE-NL. It is supplemented with several existing scales, as well as newly developed scales and items, which are relevant for employees in low-skilled jobs. The MAISE-Easy is aimed at employees in lower-skilled jobs rather than at employees with a low educational level per se. 

Six focus groups were organized with employees in low-skilled jobs in five Dutch companies from the financial, cleaning, logistic, food and industrial sectors. The number of employees in the focus groups varied from 2 to 9, and the focus group meetings lasted about two hours. Each focus group meeting consisted of two parts. The first part of the meeting was spent on asking employees in low-skilled jobs about the meaning they attached to SE. The second part focused on the MAISE-NL and was inspired by the “cognitive debriefing method” [12] meaning that, for each item of the MAISE-NL, employees were asked to actively look at the MAISE-NL and give their first impressions. Examples of questions asked were: “Is it clear and understandable?”, “Is it easy to fill in or not?”, “Do you believe other colleagues can fill it in?”, or “What kind of items concerning healthy working are you missing?”.

After the first round of the focus groups, the MAISE-NL appeared to be generally clear and understandable for employees. It seemed desirable though to rename SE “staying healthy at work” throughout the whole questionnaire to increase the comprehensibility for employees in low-skilled jobs. All items were checked for positive formulation and adjusted if necessary. 

Further, the MAISE-NL contained an area about the employees’ ideas about the meaning of SE and an area tapping the level of SE of the employees themselves. Based on the focus group, employees in low-skilled jobs did not grasp or appreciate the difference between both areas, and therefore, the first area of MAISE-NL was not included in the MAISE-Easy. 

Once the questionnaire was adapted, it was sent to the human resource (HR) manager and/or supervisors of each company who were asked to comment on the input of the employees and report on items that they were missing. A focus group was also organized with team leaders and supervisors in the cleaning company because employees of this company were very low-educated and often non-Dutch. For this reason, the questionnaire was made available both in Dutch and English. The MAISE-Easy items were translated from Dutch to English by a professional translator who is an English native speaker, has a proficiency level in Dutch and experience as a researcher. The retranslation was compared with the Dutch version of the MAISE-Easy and discussed by the developers of the questionnaire (EH and IH). Both the Dutch and English versions of the MAISE-Easy are available from the authors upon request. 

To summarize, the hypothesized version of the MAISE-Easy includes four areas: (1) level of SE (which measures the SE level of employees), (2) factors affecting SE, (3) overall responsibility for SE and (4) responsibility for factors affecting SE.

*Level of SE* (Area 1) includes five scales. SE is measured by means of two scales from the MAISE-NL: health (3 items) and productivity (6 items). Based on the focus group, three scales were added on the indication of the employees: job control (5 items) and social work climate (4 items), which were measured through items developed by the researchers, and the self-efficacy scale (5 items), which was based on the scale “effort” from the general self-efficacy scale (GSES-12) [13]. The wording of the latest scale items was adjusted. The response scale of Area 1 was modified from a 5-point Likert scale (1 = Strongly agree, 5 = Strongly disagree) to a 5-point frequency Likert scale (1 = Never, 5 = Always). Employees reported these scales as relevant to their SE and being in line with the SE component valuable work deduced from the definition of Van der Klink et al. [2,3].

*Factors affecting SE* (Area 2) includes five scales. Employees were asked which factors (e.g., more support from my manager) might be helpful (or not) to stay healthy at work and to become more sustainably employable. Three of these scales were taken from the MAISE-NL but slightly adapted; one item on clarity and one item on freedom were added to the work organization scale (9 items). The wording of the adapted work possibilities scale (4 items) was adjusted. The original lifestyle and work–life balance scales were combined into the health and lifestyle scale (9 items) in which one item on physical movement and five items on lifestyle were added based on employees’ request. Two new self-developed scales, social support (3 items) and communication and collaboration (5 items), were added based on the researcher’s interest and the focus group input. Lastly, the response scale was adapted from a 5-point Likert scale (1 = Nothing, 5 = A lot) to a 3-point scale (1 = It is fine as it is, 2 = will not help me much, 3 = will help me a lot).

*Responsibility for SE* (Area 3) includes one scale of one item: “With whom does the responsibility for sustainable employability lie according to you?”. Only the wording of this scale was adapted to” who should take responsibility for being healthy at work?”. The response scale ranged from 1 = Only my company to 5 = Only me.

*Responsibility for factors affecting SE* (Area 4) includes five scales. In total, 17 items from the original scales of the MAISE-NL were kept, and 13 new items were added. These new items measured the responsibility for the factors that were added to Area 2 of the MAISE-Easy—*Factors affecting SE*. The 5-point response scale remained the same as in the MAISE-NL (1 = Only my company, 5 = Only me).

In sum, the development process resulted in a final hypothesized set of items organized into four areas including 16 scales (see Figure 1). All items were measured from the employee’s perspective and were well aligned with the four SE core components based on the definition of Van der Klink. [2,3]. The core components of health and productivity are reflected in the “health” and “productivity” scale in Area 1 (level of SE). The core component of valuable work is reflected in the scales of social work climate, job control and self-efficacy. The core component of long-term perspective is not explicitly included in the MAISE-Easy as a scale but is implicit in one item in the productivity scale (“I have the feeling that I will be able to carry on with my job until I retire”).

### 2.2. Population, Design and Procedure

The MAISE-Easy was tested in a sample of employees who varied in gender, age and educational level. Although educational levels varied from no education to university level, all employees performed low-skilled jobs [14], and 92% of the employees in our sample had a secondary vocational education or lower. Data were collected between May and October 2019. The employees’ participation in the study was voluntary. The sample included employees from five Dutch organizations: a financial company, a cleaning company, a logistic company, a food processing company and an industrial company. The low-skilled jobs in the cleaning, logistic, food processing and industrial companies mainly consisted of physically demanding work (e.g., carrying heavy loads, standing), while the low-skilled jobs in the financial company consisted of relatively simple administrative tasks (deskwork). A total of 64% of employees fully completed the questionnaire in the industrial company, 54% in the cleaning company, 53% in the financial company, 32% in the logistic company and 12% in the food processing company. The average response rate of employees in all organizations was 53%. Table 1 provides an overview of the demographic characteristics for the total sample and the five organizations separately.

### 2.3. Measures

In addition to the MAISE-Easy items described above, items on gender, age, educational level, vitality and work engagement (i.e., for testing criterion validity) were included in the questionnaire.

Vitality was measured by means of the scale vitality of the Dutch version of the Utrecht Work Engagement Scale (UWES) (5 items) [15]. Work engagement was measured by means of the shortened Dutch version of the Utrecht Work Engagement Scale (UWES-3). The UWES-3 includes all three dimensions of work engagement (vigor, dedication, absorption). This short version of UWES-9 is proven to be reliable and valid [16]. The vocabulary of the UWES items was checked for comprehensibility and appeared to be understandable. The response scale ranged from 1 “Never” to 7 “Always/Everyday”. 

### 2.4. Data Analysis

The original MAISE-NL was used as the starting point for the development of the MAISE-Easy, but in the development process, major adaptations were made. In addition, the MAISE-Easy was specifically developed for employees in low-skilled jobs and hence had a different target group than the MAISE-NL. We consider the MAISE-Easy a new instrument, and therefore, we decided to take an integral approach to analyzing the psychometric properties of the MAISE-Easy; we first performed an exploratory factor analysis, and in the second step, we performed a confirmatory factor analysis [17].

To investigate the validity and reliability of the MAISE-Easy, several statistical analyses were performed using IBM SPSS Statistics version 26 (IBM Corp., Armonk, NY, USA). 

First, an exploratory factor analysis was performed employing a principal component analysis (PCA) with oblimin rotation to investigate constructs’ validity of the MAISE-Easy. All components extracted had an eigenvalue >1. The items that had factor loadings higher than 0.30 or lower than −0.30 on the same factor were considered highly related to each other. 

Second, a confirmatory factor analysis (CFA) was performed to further validate the MAISE-Easy areas and scales. CFA was conducted by means of JAMOVI version 0.9.5.12 [18]. JAMOVI uses the maximum likelihood estimation method, which is scale invariant. We constructed the models based on the PCA results. The exact fit of the model was assessed with the Chi-square index. Because of the high sensitivity of the Chi-square index to sample size [19], we used several comparative and parsimonious fit indices [20]: the root mean square error of approximation (RMSEA, which should be lower than 0.08); the comparative fit index (CFI) and the Tucker–Lewis index (TLI, also known as the non-normed fit index, which should both be 0.90 or higher); and the standardized root mean square residual (SRMR, which should be lower than 0.08). For some scales, we allowed residual errors of some items to correlate. 

Third, the reliability (internal consistency) of the MAISE-Easy was analyzed by means of Cronbach’s alpha calculations. The following categories were used: moderate (alpha ≤ 0.70), adequate to good (alpha ≥ 0.70 and ≤0.80) and good (alpha ≥ 0.80).

Fourth, Pearson correlation coefficients were performed to examine the criterion validity of the MAISE-Easy scales of Area 1, 3 and 4 by comparison with the criteria vitality and work engagement with the MAISE-Easy scales. With regard to the predictors of gender, age and educational level, one-way ANOVAs were performed. 

## 3. Results

### 3.1. Construct Validity and Reliability

#### 3.1.1. Level of SE

Table 2 and Table 3 show the results of the PCA (construct validity) and reliability analyses of the MAISE-Easy items of Area 1—*Level of SE*. 

*Level of SE* consists of five scales: (1a) productivity (six items), (1b) social work climate (four items), (1c) health (three items), (1d) job control (five items) and (1e) self-efficacy (five items). 

A PCA was performed for the scales of productivity, social work climate, health and job control (see Table 2). Four factors with eigenvalue > 1 (5.70, 2.31, 1.57 and 1.07) were drawn, explaining 59.14% of the total variance. The item “I enjoy my job” loaded high on productivity but was moved to the health scale because this item can be related to mental health. The item “I have the feeling I will be able to carry on with my job until I retire” scored high on productivity but the highest on the health factor. As this item is more related to being productive than being healthy, we decided to keep this item in the productivity scale. Finally, it was decided to keep the item “I can work safely (temperature, light, safe surroundings, protective equipment)” in the health scale despite a high score on social work climate, as it relates more clearly to the physical health and environment of employees.

This four-factor structure was clearly confirmed in the CFA (see Table 4). We allowed four error terms to correlate in the CFA (two within the productivity scale and two within the health scale). Cronbach’s alphas of scales 1a, 1b and 1e were adequate to good. Cronbach’s alpha of scale 1c was moderate, while it was good for scale 1d. Based on the PCA and for content reasons, scale 1c was kept as such despite a moderate Cronbach’s alpha. 

A separate PCA was performed for the self-efficacy scale (see Table 3), as it is an existing validated scale, and only minor wording changes were made. As expected, one factor with eigenvalue >1 (2.63) was drawn, explaining 52.60% of the total variance. The CFA clearly confirmed this structure (See Table 4). One error term was allowed to correlate.

#### 3.1.2. Factors Affecting SE, Responsibility for SE and Responsibility for Factors Affecting SE

Area 2—*Factors affecting SE*—was measured using a categorical response scale. Therefore, a PCA could not be performed for this area. This area can be considered on the item level, and the items are categorized based on content but should not be averaged (See Appendix A). 

Area 3—*Responsibility for SE*—was measured using only one item; therefore, the factor structure was not tested. 

Table 5 shows the results of the PCA and reliability analyses of the MAISE-Easy items of Area 4—*Responsibility for factors affecting SE*. Five factors with eigenvalue > 1 (9.24, 3.33, 1.78, 1.42 and 1.29) were drawn, explaining 56.84% of the total variance. Based on the PCA, several adjustments were made to the scales. 

Scale 4a (support) was removed. Item 2 scored highest on collaboration and was therefore moved to this new scale for content reasons. Items 1 and 3 were moved under a newly created scale: job atmosphere.

Most items of scale 4b (work organization) scored highest on work organization and remained in the scale. However, items 4 and 5, initially expected to score high on work organization, scored highest on job atmosphere. Because the content matched with the new scale, the items were moved to the job atmosphere scale.

In scale 4c (health and lifestyle), all items scored highest on the same factor, except for item 21 and 13, which scored highest on job atmosphere. Based on the content, item 21 was moved to the job atmosphere scale. However, item 13 was kept in the health and lifestyle scale due to content reasons.

No adjustments were made to adapted job possibilities (scale 4d). 

Scale 4e (communication and collaboration) was split into two new scales: collaboration and communication. The new collaboration scale included collaboration items of the initial scale. The new communication scale included the communication items of the initial scale. Communication and adapted job possibilities (scale 4d) items both scored highest on the same factor. Given the content of the items, the scales could not be combined. Therefore, six scales were kept in this section, despite the five PCA components.

The CFA showed that a six-factor structure had better fit indices than the five-factor structure. The CFA also showed the fit of this area of the MAISE-Easy improved when item 13, “More variety in physical movements during the day”, was deleted (see Table 4). Although fit indices CFI and TLI were slightly below the threshold levels, both six-factor structures were generally confirmed in the CFA (see Table 4). We allowed eight error terms to correlate in both solutions. 

Cronbach’s alpha was moderate for scale 4a, while it was adequate to good for scales 4d, 4e and 4f. For scales 4b and 4c, it was good.

Based on the PCA and CFA, the MAISE-Easy resulted in a set of items organized into four areas, including 17 scales. All items are measured from the employee’s perspective. Figure 2 provides an overview of the areas, scales and number of items per scale of the MAISE-Easy after adaptations based on the PCA analyses.

### 3.2. Criterion Validity

In this section, we focused on the criterion validity of MAISE Area 1 only. We examined the correlations of all subscales of level of SE (Area 1) with the criteria vitality and work engagement, and we performed one-way ANOVAs of the level 1 subscales with gender, age and educational level. 

Table 6 shows the Pearson correlation coefficients of the MAISE-Easy scales. For Area 1, as hypothesized (Hypothesis 2a), scales 1a, 1b, 1c and 1e, especially 1a (productivity) and 1c (health), were moderately to highly associated with both criteria of vitality and work engagement, hereby confirming criterion validity of the MAISE-Easy. 

Table 7 shows the means, standard deviations and ANOVAs of the subscales of MAISE-Easy in the total sample and the means, ranges and standard deviations for gender, age and educational level. For *Factors affecting SE* (Area 2), we did not report the mean scores per scale, as the response categories were categorical. For the scores on item level based on the chi-square test, see Table A1 in Appendix A. 

With regard to gender, we only found significant mean difference for the self-efficacy; women reported having slightly more self-efficacy than men. With regard to age, we found a significant mean difference for productivity; older employees (>45 years) reported feeling slightly more productive than younger employees, the mean difference being limited though. With regard to the education level, we found significant differences for social work climate and job control between educational levels. For social work climate, the lower the educational level, the most frequently good social work climate was reported. Employees with the lowest educational level (primary school) reported higher job control compared to the higher-educated employees, who reported having low job control. Hypothesis 2b was partially confirmed. 

## 4. Discussion

This paper describes the development and validation of the MAastricht Instrument of Sustainable Employability (MAISE-NL) adapted for employees in low-skilled jobs (MAISE-Easy). The MAISE-Easy is based on the MAISE-NL and was adapted by means of focus groups conducted among employees in low-skilled jobs. The MAISE-Easy aims to measure sustainable employability (SE) from an employee’s perspective and includes 17 scales divided over four areas: (1) level of SE (5 scales), (2) factors affecting SE (5 scales), (3) responsibility for SE (1 scale), (4) responsibility for factors affecting SE (6 scales). The MAISE-Easy construct validity (PCA and CFA) and reliability were good, confirming Hypotheses 1a and 1b. Two scales (1c and 4a) had a somewhat lower reliability, but still acceptable, and were kept for content reasons and clear factor structure (4a). Correlational analyses showed that the criterion validity of the MAISE-Easy Area 1 (level SE) with the criteria vitality and engagement was good (Hypothesis 2a was confirmed). Hypothesis 2b was partially confirmed; only some subscales of MAISE Area 1 (level of SE) varied across the subgroups. No differences were found between men and women, except for self-efficacy, which women reported slightly more than men. With regard to age, we found older employees to report being productive slightly more than younger employees. This seems in line with our expectation that productivity increases with experience [21,22]. We found no age differences for the other aspects of level of SE. Contrary to expectations, employees with the lowest educational level (primary school) scored higher on productivity, health, social work climate and job control (but still low) as compared to their higher-educated colleagues. With regard to job control, this result may be explained as follows. Autonomy and job control in low-skilled jobs can be assumed to be low. The relatively higher-educated employees in these low-skilled jobs might be more bothered by these low levels of autonomy and consequently perceive job control to be very low. 

All in all, we can conclude that the MAISE-Easy has adequate to good psychometric properties and is relevant and highly needed. Most existing questionnaires tackling work and health are developed for middle to highly educated employees rather than for employees in low-skilled jobs who often have a lower education level. Several adjustments had to be made in the MAISE-NL in order to make the questionnaire suitable for use among employees in low-skilled jobs. This indicates that the validity of the MAISE-NL in this group was limited. The MAISE-Easy may facilitate the inclusion of employees in low-skilled jobs in needs assessments and can also be used to develop and evaluate interventions, which are better aligned with the needs and circumstances of this group of employees. 

The PCA showed that some further adjustments were indicated in the MAISE-Easy. This might be the result of some items still being too ambiguous for employees in low-skilled jobs or due to the variety of employees, which may have been larger than in the focus groups. For instance, employees worked in different sectors, had different types of jobs and ethnicities. The PCA was, therefore, very valuable for further fine tuning the questionnaire to the vocabulary and work context of employees in low-skilled jobs.

### 4.1. Recommendations for Future Use of the MAISE-Easy

The MAISE-Easy will facilitate research in the field of work and health in the understudied group of employees in low-skilled jobs. The instrument will also facilitate employers in developing or selecting SE interventions tailored to the needs of the more vulnerable and underserved group of employees in low-skilled jobs. The MAISE-Easy can be used as a needs assessment to help in the development of decent and more inclusive work conditions for occupational groups that are more vulnerable to SE, such as employees in low-skilled jobs. Interventions that are better aligned to the needs of employees in low-skilled jobs will likely be more effective. The MAISE-Easy can be used as an evaluation tool after the intervention implementation as well, to evaluate whether the implemented SE intervention was effective. Based on the preferences of organizations, different work-related outcomes (such as sickness absence, presenteeism, burnout) could be added or replaced to explore more the relationship of SE with these outcomes. 

### 4.2. Methodological Reflection and Future Research

The study sample included employees working in different companies and sectors and varied in age, gender and educational level. The sample also included employees with higher educational levels, (e.g., university degree) because the inclusion criteria in this study related to having a low-skilled job rather than a low educational level. It shows that some higher-educated employees preferred to work in low-skilled jobs, which may be related to work pressure and too much autonomy in higher-skilled jobs. The average response rate was 53%, which can be considered relatively high given the target population and comparable with other organizational surveys. The MAISE-Easy was translated into English. However, employees who are illiterate or unable to read Dutch or English are not yet being served with this measure. For some employees in low-skilled jobs, the method of a questionnaire remains difficult. Moreover, the MAISE-Easy is a rather lengthy questionnaire. It might therefore be relevant to consider other methods to quantify employee perspectives on SE and ways to include this specific target group, for instance, using pictograms. Additionally, the response scale of Area 2 (factors affecting SE) has some limitations (1 = It is fine as it is, 2 = will not help me much, 3 = will help me a lot). The response scale was inserted based on the advice of employees in the focus groups, as they found that easier to understand. However, the response scale still turns out to be too ambiguous for the understanding of the respondents. This raised some difficulties in the analyses and interpretation of results. Future adaptations of the response scale may be helpful for future use of the MAISE-Easy as a needs assessment among employees in low-skilled jobs. Finally, the results may be influenced by some forms of common method variance or artificial inflation of synchrony in the answers, which is inherent to all self-reported and cross-sectional data [23]. With regard to the criterion validity, we could not infer causality, and future studies with a longitudinal design are needed. Translations of the MAISE-Easy into several immigrant employees’ native languages should also be considered to increase the internal and external validity (i.e., transferability).

## 5. Conclusions

The MAISE-Easy is a valid adaptation of the MAISE-NL for an underserved group of employees that is often ignored in research. Very few survey instruments have been tested regarding their feasibility for employees in low-skilled jobs and even fewer were optimally adapted. Our new instrument was adapted using both focus group sessions with the target group and robust psychometric methods. The MAISE-Easy thus appears to be a reliable and valid measurement instrument for measuring aspects of sustainable employability in employees who work in low-skilled jobs. The MAISE-Easy includes scales to evaluate the employee perspective on the level of SE, factors affecting SE, responsibility for SE, responsibility for factors affecting SE and vitality and work engagement. We recommend for researchers to use this instrument for SE studies and employers to use the MAISE-Easy as a needs assessment for developing SE interventions that will be more readily accepted and more effective for employees in low-skilled jobs. 

## Figures and Tables

**Figure 1 ijerph-19-07977-f001:**
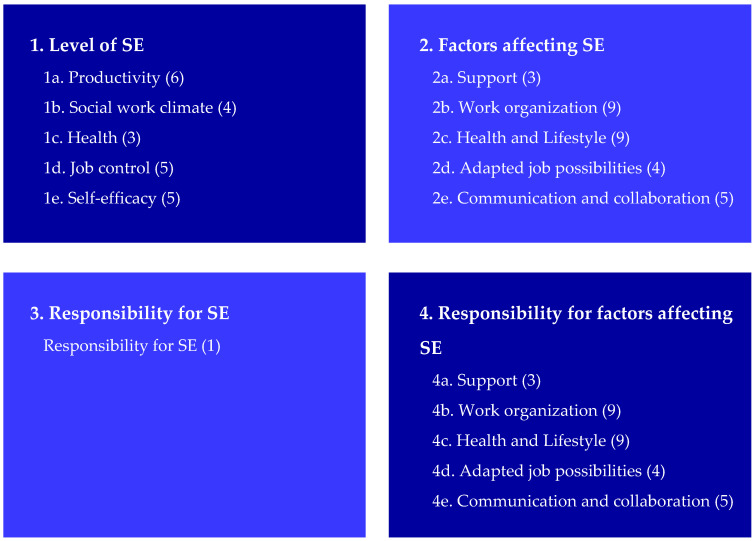
Areas, scales and number of items per scale of the MAastricht Instrument of Sustainable Employability (MAISE-NL) adapted for employees in low-skilled jobs (MAISE-Easy).

**Figure 2 ijerph-19-07977-f002:**
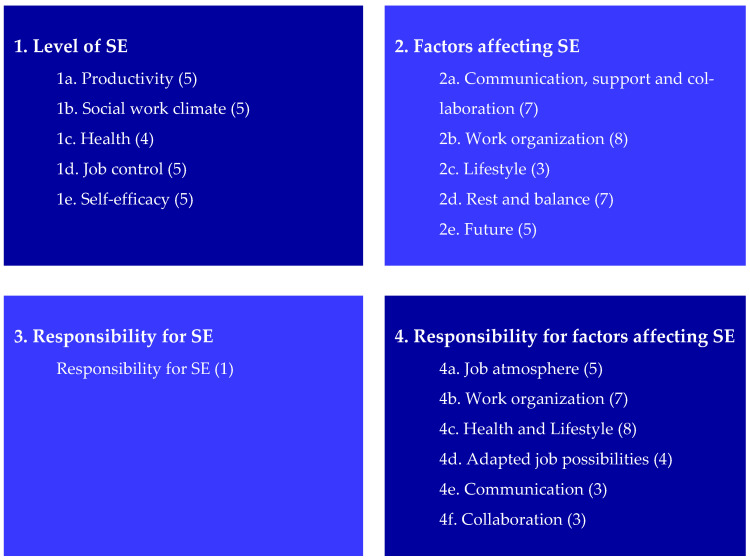
Areas, scales and number of items per scale of the MAISE-Easy after adaptations based on PCA.

**Table 1 ijerph-19-07977-t001:** Sample characteristics: number of employees, age (mean and %), gender (%) and educational level (%).

Variable	Total	Company				
		Financial	Cleaning	Logistic	Food	Industrial
N (range)	1054–1084	118–120	118–132	11	46	761–775
Age (mean)	43.3	52.9	41.0	35.8	49.7	41.9
≤45 (%)	48.3	19.5	62.7	72.7	17.4	51.8
>45 (%)	51.7	80.5	37.3	27.3	82.6	48.2
Gender (%)						
- men		38.3	22.0	72.7	87.0	87.2
- women	26.3	61.7	78.0	27.3	13.0	12.8
Educational level (%)						
- PS/Did not finish school	9.0	0.0	21.2	9.1	0.0	9.1
- LSE, SSE, SVE 1, SVE 2	51.4	51.3	62.7	63.6	82.6	47.6
- SVE 3–4	31.6	33.6	8.5	9.1	13.0	36.3
- HPE, University	8.1	15.1	7.6	18.2	4.3	7.1

Note. PS = Primary School, LSE = Lower Secondary Education, SSE = Senior Secondary Education, SVE = Secondary Vocational Education, HPE = Higher Professional Education.

**Table 2 ijerph-19-07977-t002:** PCA MAISE-Easy Area 1—Level of SE (productivity, social work climate, health, job control), oblimin rotation.

	How Do You Feel about Your Job?				
#	Item	Productivity	Climate	Health	Control
1	I have the knowledge to be able to do my job well	**0.807**	−0.075	−0.079	−0.017
3	I do my job efficiently	**0.862**	−0.009	−0.082	−0.076
4	I have the feeling that the job I do is useful	**0.598**	0.165	0.110	0.099
5	I have the feeling that I will be able to carry on with my job until I retire	**0.324**	0.081	0.454	0.208
6	I am productive when I am working	**0.765**	0.014	0.007	−0.010
	Cronbach’s alpha scale 1a productivity	0.742			
7	I feel safe and secure when I am at work	0.208	**0.595**	0.161	−0.038
8	I get help and support at work	−0.061	**0.822**	−0.012	−0.026
9	I am treated with respect at work	−0.018	**0.825**	0.073	−0.030
10	I feel appreciated/get compliments at work	0.009	**0.722**	−0.114	0.187
	Cronbach’s alpha scale 1b social work climate		0.794		
2	I enjoy my job	0.357	0.312	**0.329**	0.148
11	I can work safely (temperature, light, safe surroundings, protective equipment)	0.137	0.459	**0.198**	0.016
12	I get physical complaints (pain) due to my job (R)	−0.044	−0.051	**0.819**	0.076
13	My job is stressful (R)	−0.090	0.137	**0.693**	−0.138
	Cronbach’s alpha scale 1c health			0.624	
14	I have a say in what happens at work	0.037	0.155	−0.137	**0.762**
15	I can decide the type of work I do	0.032	0.077	−0.056	**0.821**
16	I have seen my ideas put into practice in my workplace	−0.049	0.240	−0.107	**0.678**
17	I can decide how to organize my work	0.012	−0.170	0.148	**0.800**
18	I can take a break when I think it is necessary	−0.040	−0.076	0.053	**0.649**
	Cronbach’s alpha scale 1d job control				0.813

Note. Climate = Social work climate, Control = Job control, (R) = recoded items. The bold numbers indicate the chosen scale for each item.

**Table 3 ijerph-19-07977-t003:** Principal component analysis (PCA), MAISE-Easy Area 1—Level of SE (self-efficacy).

	How Do You Feel about Your Job?	
#	Item	Self-Efficacy
19	When I have something unpleasant to do, I stick to it until I finish it	**0.620**
20	When I decide to do something, I go right to work on it	**0.721**
21	If I can’t do a job the first time, I keep trying until I can	**0.815**
22	Failure just makes me try harder	**0.728**
23	When I make plans, I am certain I can make them work	**0.730**
	Cronbach’s alpha scale 1e self-efficacy	0.766

Note. The bold numbers indicate the chosen scale for each item.

**Table 4 ijerph-19-07977-t004:** Fit indices of the MAISE-Easy areas.

		Chi-2 (df)	CFI	TLI	SRMR	RMSEA
1	Level of SE (four factors)	665 (141) **	0.925	0.909	0.071	0.058
1	Level of SE (self-efficacy)	33.7 (4) **	0.978	0.946	0.023	0.083
4	Responsibility for factors affecting SE (six factors)	2276 (377) **	0.866	0.846	0.068	0.069
4	Responsibility for factors affecting SE (six factors, without item #13)	1902 (354) **	0.887	0.871	0.053	0.064

Note. CFI = Comparative Fit Index; TLI = Tucker–Lewis Index; SRMR = Standardized Root Mean Square Residual; RMSEA = Root Mean Square Error of Approximation. ** *p* < 0.01.

**Table 5 ijerph-19-07977-t005:** PCA MAISE-Easy Area 4—Responsibility for factors affecting SE, oblimin rotation.

	Who Do You Think Should Be Responsible for the Changes Mentioned Below?					
#	Item	Job Atmosphere	Work Organization	Health and Lifestyle	Adapted Job Possibilities/Communication	Collaboration
1	Getting more support from my direct manager	**0.528**	0.101	0.001	−0.057	0.284
3	Getting complimented at work more often than I do now	**0.582**	0.179	−0.029	0.034	0.041
4	Improving the atmosphere within my department/shift/team (respect, openness, motivation)	**0.556**	0.116	0.055	0.128	0.271
5	Improving the working conditions (noise, temperature, protective equipment)	**0.589**	0.68	−0.007	−0.234	−0.199
21	Less pressure at work	**0.462**	−0.043	0.258	−0.333	−0.082
	Cronbach’s alpha scale 4a job atmosphere	0.691				
6	Getting opportunities to learn new things/tasks	0.321	**0.450**	0.027	−0.048	−0.026
7	Getting more variation in the type of work I do	0.107	**0.720**	−0.026	−0.066	0.002
8	Getting more challenges in the type of work I do	0.065	**0.821**	0.004	0.028	−0.035
9	Using my knowledge/skills at my place of work better	−0.052	**0.816**	0.091	0.080	0.061
10	To be given more responsibility at my place of work	−0.085	**0.838**	0.057	0.040	0.049
11	To be given more freedom in how I do my job	0.028	**0.727**	−0.017	−0.119	0.013
12	Getting more clarity about my task/work	0.196	**0.508**	−0.020	−0.204	−0.035
	Cronbach’s alpha scale 4b work organization		0.883			
13	More variety in physical movements during the day (lifting, bending, repetitive movement)	0.371	0.153	**0.295**	−0.144	−0.052
14	More time to take exercise	−0.084	0.143	**0.697**	−0.100	−0.077
15	Reach a healthy weight	−0.215	0.099	**0.763**	0.093	0.094
16	Eating healthily at work	0.053	0.041	**0.671**	−0.026	−0.043
17	Getting enough rest after work	0.015	−0.034	**0.837**	0.027	0.004
18	Improving how I sleep	−0.010	−0.066	**0.829**	0.029	0.065
19	A better balance between my work and private life	0.074	−0.044	**0.740**	−0.072	0.015
20	Learning to manage stress better	0.122	−0.054	**0.655**	0.028	0.084
	Cronbach’s alpha scale 4c health and lifestyle			0.864		
22	Introduce more flexibility into my working hours/schedule	0.068	0.052	0.126	**−0.707**	−0.139
23	More attention to career development	0.052	0.299	0.035	**−0.531**	0.024
24	Working fewer hours per week	−0.098	−0.008	0.145	**−0.714**	−0.013
25	Changing my tasks/job	−0.052	0.329	−0.068	**−0.572**	0.116
	Cronbach’s alpha scale 4d adapted job possibilities				0.791	
26	Having more say in things that I am concerned with at work	−0.091	0.300	−0.053	**−0.499**	0.267
27	Better communication about the day-to-day running of the company	0.228	0.015	−0.087	**−0.634**	0.081
28	More clarity about who I should speak to if I have problems	0.180	−0.071	−0.024	**−0.534**	0.312
	Cronbach’s alpha scale 4e communication				0.719	
2	Getting more support from my direct colleagues	0.288	0.096	0.057	0.233	**0.601**
29	Better cooperation/interaction with my colleagues	−0.144	0.073	0.109	−0.106	**0.804**
30	Better cooperation/interaction with my direct manager	0.009	−0.009	0.111	−0.277	**0.720**
	Cronbach’s alpha scale 4f collaboration					0.729

Note. The bold numbers indicate the chosen scale for each item.

**Table 6 ijerph-19-07977-t006:** Pearson correlations of MAISE-Easy scales and items (N ranges from 1033 to 1076).

#	Variable ^a^	1a	1b	1c	1d	1e	3	4a	4b	4c	4d	4e	4f	5
	*MAISE-EASY Scales*													
1a	Productivity	-												
1b	Social work climate	0.51 **	-											
1c	Health	0.54 **	0.60 **	-										
1d	Job control	0.28 **	0.38 **	0.24 **	-									
1e	Self-efficacy	0.42 **	0.27 **	0.23 **	0.14 **	-								
3	Overall responsibility for SE	0.13 **	0.15 **	0.22 **	0.09 **	0.07 *	-							
4a	Atm.-res	0.19 **	0.35 **	0.32 **	0.29 **	0.04	0.29 **	-						
4b	Org.-res	0.26 **	0.32 **	0.24 **	0.34 **	0.10 **	0.21 **	0.61 **	-					
4c	H and L-res	0.18 **	0.26 **	0.28 **	0.12 **	0.11 **	0.21 **	0.37 **	0.35 **	-				
4d	Adap.res	0.20 **	0.22 **	0.17 **	0.29 **	0.02	0.17 **	0.52 **	0.59 **	0.37 **	-			
4e	Com.-res	0.18 **	0.28 **	0.17 **	0.28 **	0.03	0.19 **	0.54 **	0.57 **	0.29 **	0.65 **	-		
4f	Coll.-res	0.13 **	0.25 **	0.16 **	0.14 **	0.13 **	0.22 **	0.41 **	0.45 **	0.37 **	0.33 **	0.43 **	-	
5	*Criteria*													
	Vitality	0.49 **	0.39 **	0.46 **	0.23 **	0.40 **	0.20 **	0.25 **	0.25 **	0.27 **	0.21 **	0.22 **	0.18 **	-
	Work engagement	0.56 **	0.42 **	0.47 **	0.29 **	0.38 **	0.17 **	0.24 **	0.27 **	0.23 **	0.22 **	0.21 **	0.16 **	0.85 **

Note. * *p* < 0.05 level, ** *p* < 0.01. **^a^** Explanation of variable names: Com., sup., and coll. = Communication, support and collaboration, Atm.-res = Responsibility for job atmosphere, Org.-res = Responsibility for work organization, H and L-res = Responsibility for health and lifestyle, Adap.res = Responsibility for adapted job possibilities, Com.-res = Responsibility for communication, Coll.-res = Responsibility for collaboration.

**Table 7 ijerph-19-07977-t007:** Means (M), standard deviations (SD) and percentiles of the MAISE-Easy scales for the total sample and subgroups (gender, age, educational level) for Areas (1) level of SE, (3) responsibility for SE, (4) responsibility for factors affecting SE and for the criteria (vitality and work engagement).

Scale/Proxies	M (Range)	SD	25th Perc.	75th Perc.	M (Range)	SD	M (Range)	SD	M (Range)	SD	M (Range)	SD	M (Range)	SD	M (Range)	SD	M (Range)	SD	M (Range)	SD
	Total Sample (*n* = 1035–1076)				Men (*n* = 774–796)		Women (*n* = 261–280)		≤45 (*n* = 493–511)		>45 (*n* = 531–552)		Primary School/ Did not Finish School (*n* = 92–95)		LSE, SSE, SVE 1 and SVE 2 (*n* = 528–540)		SVE 3–4 (*n* = 329–333)		HPE and University (*n* = 83–85)	
**1. Level of SE**																				
1a. Productivity	3.97 (1.4–5)	0.68	3.60	4.40	3.97 (1.6–5)	0.67	3.97 (1.4–5)	0.69	3.89 (1.6–5)	0.66	4.05 (1.4–5)	0.68	4.04 (1.6–5)	0.78	4.00 (1.6–5)	0.68	3.90 (1.4–5)	0.63	3.94 (2.2–5)	0.65
1b. Social Work Climate	3.62 (1–5)	0.82	3.00	4.25	3.61 (1–5)	0.82	3.64 (1–5)	0.85	3.59 (1–5)	0.84	3.64 (1–5)	0.81	3.93 (2–5)	0.84	3.61 (1–5)	0.83	3.57 (1–5)	0.78	3.42 (1–5)	0.86
1c. Health	3.76 (1–5)	0.71	3.25	4.25	3.76 (1–5)	0.71	3.75 (1.5–5)	0.70	3.75 (1.25–5)	0.71	3.77 (1–5)	0.70	3.90 (1.25–5)	0.86	3.75 (1.25–5)	0.70	3.73 (1.25–5)	0.62	3.74 (1–5)	0.77
1d. Job Control	2.58 (1–5)	0.89	1.80	3.20	2.56 (1–5)	0.87	2.62 (1–5)	0.92	2.54 (1–5)	0.91	2.61 (1–5)	0.87	2.75 (1–5)	0.95	2.51 (1–5)	0.87	2.65 (1–5)	0.88	2.55 (1–5)	0.95
1e. Self-efficacy	4.07 (1–5)	0.71	3.60	4.60	4.04 (1–5)	0.71	4.17 (1.8–5)	0.70	4.08 (1.2–5)	0.68	4.06 (1–5)	0.73	3.95 (1–5)	0.89	4.10 (1.4–5)	0.70	4.10 (1.2–5)	0.64	4.02 (2–5)	0.73
**3. Responsibility for SE**	2.83 (1–5)	0.64	3.00	3.00	2.81 (1–5)	0.64	2.91 (1–5)	0.62	2.83 (1–5)	0.69	2.83 (1–5)	0.59	2.88 (1–5)	0.96	2.83 (1–5)	0.58	2.81 (1–5)	0.62	2.88 (1–5)	0.62
**4. Responsibility for factors affecting SE**																				
4a. Job atmosphere	2.43 (1–5)	0.61	2.00	2.80	2.40 (1–5)	0.61	2.50 (1–5)	0.62	2.39 (1–5)	0.64	2.47 (1–5)	0.59	2.62 (1–5)	0.77	2.46 (1–5)	0.63	2.33 (1–4.2)	0.54	2.34 (1–3.6)	0.57
4b. Work Organization	2.67 (1–5)	0.72	2.29	3.00	2.64 (1–5)	0.70	2.74 (1–5)	0.77	2.54 (1–5)	0.70	2.78 (1–5)	0.73	2.82 (1–5)	0.85	2.72 (1–5)	0.73	2.56 (1–5)	0.66	2.54 (1–4.14)	0.69
4c. Health and Lifestyle	3.74 (1–5)	0.72	3.25	4.25	3.66 (1–5)	0.73	3.95 (1–5)	0.66	3.67 (1–5)	0.75	3.80 (1–5)	0.69	3.94 (1–5)	0.73	3.79 (1–5)	0.70	3.59 (1–5)	0.73	3.68 (1.5–5)	0.72
4d. Adapted Job Possibilities	2.50 (1–5)	0.81	2.00	3.00	2.41 (1–5)	0.77	2.77 (1–5)	0.85	2.35 (1–5)	0.80	2.63 (1–5)	0.79	2.74 (1–5)	0.82	2.59 (1–5)	0.82	2.32 (1–4.5)	0.75	2.36 (1–4)	0.72
4e. Communication	2.39 (1–5)	0.72	2.00	3.00	2.35 (1–5)	0.69	2.51 (1–5)	0.79	2.33 (1–5)	0.75	2.45 (1–5)	0.69	2.60 (1–5)	0.79	2.46 (1–5)	0.75	2.24 (1–5)	0.64	2.29 (1–3.33)	0.60
4f. Collaboration	3.07 (1–5)	0.73	2.67	3.67	3.05 (1–5)	0.74	3.13 (1–5)	0.69	3.05 (1–5)	0.69	3.09 (1–5)	0.76	3.21 (1–5)	0.80	3.11 (1–5)	0.75	3.02 (1–5)	0.67	2.91 (1–4.33)	0.63
**Vitality and work engagement**																				
Vitality	5.14 (1–7)	1.27	4.20	6.00	5.14 (1–7)	1.29	5.13 (1.6–7)	1.22	5.05 (1–7)	1.21	5.21 (1–7)	1.33	5.19 (1–7)	1.42	5.20 (1–7)	1.27	5.07 (1–7)	1.23	5.00 (2–7)	1.21
Work engagement	5.17 (1–7)	1.40	4.33	6.33	5.16 (1–7)	1.43	5.20 (1.33–7)	1.32	5.09 (1–7)	1.34	5.24 (1–7)	1.47	5.31 (1–7)	1.50	5.24 (1–7)	1.38	5.08 (1–7)	1.40	4.89 (1–7)	1.48

Note. LSE = Lower Secondary Education, SSE = Senior Secondary Education, SVE = Secondary Vocational Education, HPE = Higher Professional Education. Note. Explanation of the scale scores: Scale 1 (1 = Never, 2 = Sometimes, 3 = Regularly, 4 = Often, 5 = Always), Scales 3 and 4 (1 = Only my company, 2 = Mostly my company, 3 = Both my company and myself, 4 = Mostly me, 5 = Only me.

## Data Availability

The data presented in this study are available on request from the corresponding author. The data are not publicly available due to the personal and sensitive information from the involved organizations and their participants (employer representatives and employees). The data might be traced back to the organizations and individual respondents.

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
