# Peer review of "Validation of the MAastricht Instrument of Sustainable Employability (MAISE-NL) Adapted for Employees in Low-Skilled Jobs (MAISE-Easy)"

_ijerph, 2022, doi:10.3390/ijerph19137977_

Round 1
Reviewer 1 Report
In this article the development and validation of a new version of the Maastricht Instrument for Sustainable Employability (MAISE-NL) for low-skilled jobs was described. I think it is an important enterprise to construct good instruments for measuring sustainable employability (SE) especially focused on low-skilled jobs and workers with a lower level of education or reading and writing skills.
I like the general approach followed, where focus groups with employees in low-skilled jobs in different branches of industry discuss the topic of SE, and the way it is measured in MAISE, and adapt the scales used accordingly.
However, I was rather surprised and not convinced of the resulting scale. I looked up the original MAISE-NL instrument, and there the level of SE is measured with a scale for productivity and a scale for health issues. Although I regard this as a quite limited operationalization of the construct, it does not make sense to me to add three scales concerning job resources (job control, social work climate) and personal resources (self-efficacy). I think it is unsuitable to regard these resources as aspects of sustainable employability. That way determinants of SE contaminate measurement of the level of SE. I therefore recommend that the level of SE would be restricted to the outcome variables productivity and health.
That said, it remains unclear to me why MAISE does not use validated scales for productivity and health, but includes items that tap into different constructs such as workability (item 1, item 5), meaningful work (item 4), job satisfaction (item 2) and job safety (item 11). In my opinion the used scales lack focus and rigor.
In the introduction I would like to see a more comprehensive discussion of what SE entails, and why MAISE focuses on health and productivity primarily, rather than on alternative operationalizations of SE (e.g., workability, vitality and employability; or (work)ability, motivation and opportunity). Moreover, it remains unclear to me how (whether?) the four main components of SE are included in the MAISE-Easy. This should be better explained.
More detailed comments
Introduction
· I think the present study does not need any hypotheses. Some of the stated hypotheses are self-evident (e.g., positive correlation between SE and work engagement and vitality), not substantiated (e.g., relationship gender and educational level with productivity), or null-hypotheses (e.g., all subgroups score similar).
· A Cronbach’s alpha is generally considered adequate when it is higher than .70 rather than .60.
Materials and methods
· I do not understand what the main question was for the factors affecting SE (area 2), nor how to interpret the new answer scales. Table 5 suggests that employees are asked whether they lack certain aspects in their jobs that need to be fulfilled in order to do their work well (or become more healthy, become more sustainable employable)? Please provide this main question and explain the answer scales.
· Please give the wording of the single item on responsibility for SE (area 3)
· I would like more information on the sample in 2.2. Especially the kind of low-skilled jobs that are included in the study.
· Table 1: please be consistent in the number of decimals.
· 2.4: vitality is one of the three components of work engagement (and measured with the work engagement scale). Why wasn’t the 9-item UWES measured, so the relationships with all three components of work engagement could be assessed? It does not make sense to me to focus on one aspect and on the whole construct of work engagement.
· Moreover, I would be interested in the relationships of SE with many different work outcomes (e.g., sickness absence, presenteeism, OCB, task performance, work motivation, burnout, etc.). Why was only work engagement included?
· Gender, age and educational level are not “criteria” (page 6, line 236) but rather predictors of SE.
Results
· Table 3: when only one factor is extracted, there is no rotation of factors
· The factors affecting SE and the responsibility for these factors concern different questions about the same items. In order to be able to interpret the responsibility for certain constructs that may influence SE it is important that both areas have the same factor structure. It seems rather dysfunctional to rearrange items over different scales in area 2 and area 4. I strongly suggest that the autors test in a CFA whether the same distribution of items over scales can be used for both areas.
· Limited descriptive information is given for items of the level of SE (area 1), and no such information is given for factors affecting SE (area 2), on who is regarded as responsible for SE (area 3) and for the factors influencing SE (area 4). I would like to see a table with such descriptives (M, SD), perhaps added to Tables 2, 5 and 6.
· I would also like to see a table with correlations between all constructs in the MAISE-Easy, including all areas.
References
· Please correct reference 17. European is not the given name of Commission.
Reviewer 2 Report
Brief Summary:
The paper is very well structured and clearly and concisely written. It covers an innovative range of topics, especially by focusing on employees in low-skilled jobs. Accordingly, only minor points are criticised.
1) Introduction:
Chapter 1.1 Aim of the study, research questions and hypotheses
In this sub-chapter, the hypotheses about the goodness of fit of the measures are derived. The derivation of the hypotheses should be formulated in more detail here. In this part, it should become more clear how the authors arrive at the theoretical assumptions and the resulting hypotheses.
2) Materials and Methods
Chapter 2.3 Ethical issues: This sub-chapter should better be placed in the appendix or mentioned in a footnote, but not listed as a single sub-chapter.
Minor point:
Page 4: Box “Factors affecting SE”: somewhat unclear design because the "(5)" has slipped down after 2e. Communication and collaboration.
Page 4: Box “Responsibility for factors affecting SE”: somewhat unclear design because the "(5)" has slipped down after 2e. Communication and collaboration.
Lines 214-215: „using IBM SPSS Statistics version 26 (IBM Corp., Armonk, N.Y., 214 USA).“ That is an irrelevant information. Is not relevant for reader which statistical programme is used.
3) Results
Minor points:
False Page Numbers: There are wrong page number from page 11 on for the rest of the paper.
Round 2
Reviewer 1 Report
I appreciate the efforts of the authors to revise their manuscript and their thoughtful response to my comments on the previous version. I think the authors did a good job in clarifying some of the issues I mentioned. However, my main concerns about the approach taken by the authors remain. I do not think it is necessary that I repeat my earlier comments.
That said, I do not think we need to agree on the best way to measure sustainable employability in order for me to accept this article as a valuable contribution to the literature. I therefore think the article is almost ready for publication.
However, there is one major aspect that bothers me. That regards the measurement level of Area 2: Factors affecting SE. I now better understand what is asked here, and I realize that the measurement level of the three answer categories is categorical. This is clear to the authors as well, as they explicitly mention this in lines 419-421 (page 14), and as is evident from their inclusion of Table A1 in the Appendix. However, this also means that the principal component analysis in Table 5, and the correlations in Table 7 are unsuitable.
Actually, the questions in Area 2 are a combination of two nested questions: (1) Is this factor good or not good? And (2) if this factor is not good, would improvement of this factor help to stay healthy at work or not? It is not possible to combine the answers to this question into one ordinal scale from 1 to 3, so parametric analyses (PCA, correlation) are unsuitable. The way out would be to use two dummy variables per item, and make two separate scales or indices per group of factors, i.e., (1) the number of ‘good’ (or ‘bad’) factors, and (2) the number of factors where improvement would help to stay healthy. These scales or indices (or perhaps only the second one) could be used for correlations, and could also be analyzed in a CFA using structural equation modeling techniques for dichotomous data. I strongly advise the authors to revise their manuscript accordingly.
I would like to repeat one earlier comment: it does not make sense to hypothesize that all subgroups score similar on social work climate and self-efficacy (line 106-107). Such a null hypothesis can never be confirmed, but only rejected. I would advise the authors to reformulate this.
